# A Brief Review of In Vitro Models for Injury and Regeneration in the Peripheral Nervous System

**DOI:** 10.3390/ijms23020816

**Published:** 2022-01-13

**Authors:** Parvathi Varier, Gayathri Raju, Pallavi Madhusudanan, Chinnu Jerard, Sahadev A. Shankarappa

**Affiliations:** Centre for Nanosciences & Molecular Medicine, Amrita Institute of Medical Sciences and Research Center, Amrita Vishwa Vidyapeetham, Kochi 682041, India; parvathivarier2@gmail.com (P.V.); gayathri.r92@gmail.com (G.R.); pallavi.madhusudan@gmail.com (P.M.); chinnujerard93@gmail.com (C.J.)

**Keywords:** axon, compression, transection, microfluidic, nerve injury

## Abstract

Nerve axonal injury and associated cellular mechanisms leading to peripheral nerve damage are important topics of research necessary for reducing disability and enhancing quality of life. Model systems that mimic the biological changes that occur during human nerve injury are crucial for the identification of cellular responses, screening of novel therapeutic molecules, and design of neural regeneration strategies. In addition to in vivo and mathematical models, in vitro axonal injury models provide a simple, robust, and reductionist platform to partially understand nerve injury pathogenesis and regeneration. In recent years, there have been several advances related to in vitro techniques that focus on the utilization of custom-fabricated cell culture chambers, microfluidic chamber systems, and injury techniques such as laser ablation and axonal stretching. These developments seem to reflect a gradual and natural progression towards understanding molecular and signaling events at an individual axon and neuronal-soma level. In this review, we attempt to categorize and discuss various in vitro models of injury relevant to the peripheral nervous system and highlight their strengths, weaknesses, and opportunities. Such models will help to recreate the post-injury microenvironment and aid in the development of therapeutic strategies that can accelerate nerve repair.

## 1. Introduction

Mammalian axons in the peripheral nervous system (PNS), unlike their counterparts in the central nervous system (CNS), possess the ability to repair and regenerate to a large extent [1]. In fact, the reason for this observed dichotomy has been the focus of several studies for over a century. However, it is now generally accepted that the microenvironment within the PNS allows cellular processes that favor axonal growth [2,3,4], while that in the CNS is inhibitory [5,6]. Despite our understanding of the PNS, changes that occur during nerve injuries are not completely clear and require many in-depth studies.

Peripheral nerve injuries (PNI) are a frequent and major cause of human morbidity worldwide. Injury to the peripheral nerves results in pain and loss of function and can severely affect the patient’s mental health [7,8]. Additionally, affected patients also have to bear the financial burden of treatment and rehabilitation, thus resulting in a loss of workdays and overall quality of life [8,9]. Currently, the existing line of treatment for partial to complete PNI focuses on the management of specific symptoms, surgical repair [10,11], or complete nerve transplant [7,12]. However, newer strategies that are more practical and cost-effective in managing nerve injuries are much needed. The development of such innovative treatment options requires a thorough understanding of the cellular and molecular mechanisms involved in peripheral nerve repair and healing. In general, isolated cellular systems and animal models have proven to be extremely useful in understanding PNI-associated molecular mechanisms, electrophysiological alterations [13,14] and changes in gene expression. Importantly, in addition to biological information, inputs from experimental systems have immensely contributed to the development of mathematical and computational models to predict neural changes in PNI [15,16].

Amongst the various experimental strategies to address PNI, well-designed in vitro models, in particular, have been crucial in deciphering, discovering, and identifying many of the molecular determinants and events of nerve repair [17]. In this review, we attempt to compile, list, and expand on the existing in vitro models of nerve injury and describe the strengths and weaknesses of each model system. For in vivo and mathematical models of axonal injury, readers are directed towards other excellent reviews [7,18,19] for a better understanding of their use and application.

### 1.1. In Vitro Nerve Injury: A Window to Observe Cellular Mechanisms

In vitro nerve injury models provide vast opportunities to study cellular responses due to injury, including mechanisms for regeneration, plasticity, and degeneration. Models that mimic axotomy, drug-, or metabolite-induced nerve damage are quite useful in understanding molecular mechanisms and screening potentially beneficial therapeutic strategies. In addition to mechanistic details, it is now well-recognized that the primary initiator of injury or the type of injury plays a significant role in dictating the direction of downstream injury-associated signaling events [20,21]. Hence, it becomes quite vital that there is a thorough understanding of each model system in terms of what they address and, more importantly, what they do not.

### 1.2. PNI-Associated Cellular Changes Observed In Vitro

Generally, injury-induced interruption of axonal continuity from the neuronal cell soma is known to induce certain well-characterized cellular changes, both in the distal and proximal axonal segments. These changes have been observed in both in vivo and in vitro models. One such commonly observed cellular event is the progressive degeneration of the injured distal axonal segment, called Wallerian degeneration (WD) [22]. Sciatic nerve explants maintained for more than 2 weeks in vitro have been found to mimic several in vivo observations, such as upregulation of repair program protein c-Jun [22,23], downregulation of several myelin protein genes [22,24], and disorganization of the myelin structure around the axon [22,25,26]. In addition, WD in vitro models have been used to understand the role of important injury-induced proteins belonging to the family of calcium-dependent non-lysosomal cysteine proteases [27,28,29] such as calpains. Specifically, in vitro models such as the neurite transection models and the dorsal root ganglion (DRG) explant models have been used to demonstrate the calpain-mediated proteolysis of the neuronal cytoskeleton [30,31].

Another important hallmark of lesioned axons is the formation of retraction bulbs within the length of the axons [32,33]. Retraction bulbs lack microtubule organization and are generally considered the non-growing counterparts of neuronal growth cones [34]. In vitro models have been consistently utilized to study retraction bulb formation and the subsequent inhibition of axonal growth in the presence of inhibitors such as nocodazole [33,35,36]. Other morphological features that are consistently observed in axonal injury, including the dissipation of chromatin [37], chromatolysis [38], endoplasmic blebbing [39], and loss of dendritic spines [40], have been widely demonstrated in various in vitro models [17]. In addition to mimicking injury-induced morphological changes, in vitro models have also been successfully utilized to study the injury-induced influx of Ca^2+^ ions in axons [41,42] and associated downstream events [43,44]. Overall, it is quite apparent that in vitro models can significantly contribute to our understanding of various neuronal events, both during and after injury. Additionally, appropriate adoption of the PNI in vitro models would also contribute to the reduction, refinements, and replacement (the ‘3R’ principle) of animals in nerve injury research. Below, we list and describe the features, advantages, and disadvantages of in vitro PNI models that are currently in use (Figure 1).

## 2. Transection Injury Models

Axonal transection, or the ‘acute axotomy model’, is one of the simplest but most effective in vitro models used to investigate the process of axonal regeneration and response to injury. Robert Campenot presented the first in vitro compartment model in 1977 [45] using a three-chamber culture system in which neurites sprouting from the soma of sympathetic neurons were observed to cross a fluid-impermeable barrier. Typically, in this model, dissociated primary neurons are cultured on coated cell culture plates within compartmentalized areas. Uniformly placed scratches on the coated culture plate ensure that the regenerating neurites grow linearly on the coated areas in-between scratches, with minimal bending (Figure 2). Notably, the rake for applying scratches is made by cementing 20 insect pins placed 200 μm apart, which yields a set of 20 parallel scratches on coated cell culture plates [45,46,47].

### 2.1. The Campenot Chamber Models

The Campenot chamber allows the spatial segregation of neuronal soma and its neurites and facilitates studies related to neuronal and axonal biology [36,45,46,48]. The classical Campenot chamber is a three-compartment culture system that allows the extension of neurites from the central compartment to the two flanking peripheral compartments. A fluidic seal is obtained by the application of a hydrophobic gel such as sterilized silicone grease [46,48,49] between the walls of the compartment and the underlying culture plate surface. The silicone grease allows neurites to penetrate across while providing options to maintain varying media conditions between compartments [45,48,49]. Within the chamber, transectional axonal injury infliction has been attempted by repeatedly spraying cold sterile water through a 0.22-gauge needle into chambers that contain axons [50,51,52]. This process results in the transection of axons that can be easily visualized using conventional immunostaining methods under epifluorescence or confocal microscopes. Studies performed in this model have demonstrated that transection injury affects the rate of axonal elongation and survival [52], as well as inducing a considerable reduction in the ‘slow axonal transport’ mechanism, which is crucial for supplying cytoskeletal proteins [50] to the neuron.

### 2.2. Microfluidic Chamber Models

Injury-induced axonal changes similar to those described above have also been observed in other compartmental models, such as the 2D- and 3D microfluidic chip systems [53,54]. These model systems fabricated from an inert polymer material such as poly(dimethylsiloxane) [55,56] comprise compartments and microgrooves of varying design, which spatially separate neuronal axons and soma [57]. Axons within these microfluidic devices extend into dedicated microgrooves, which allows the selective manipulation of axons and provides opportunities for the measurement of neurite growth response [58], somal transcription activity [56], and axonal degeneration [59]. In addition, these model systems have also been utilized to understand neuronal injury associated with amyotrophic lateral sclerosis [60], the overall dynamics of the neural network, and applications involving axonal biology, drug discovery, and regenerative medicine [61].

### 2.3. Forceful Trituration

In conjunction with the microfluidic model, various types of axonal injury strategies have been applied. For instance, aspiration of a small volume of fluid using a fine-tip glass pipette placed at the entrance of the axonal compartment resulted in injury-associated upregulation of the Na_v_1.6 sodium channel [62]. Similarly, microfluidic vacuum aspiration followed by introducing air bubbles within the axonal compartment resulted in features linked to acute axonal degeneration, delayed bulb formation, and impairment of axonal mitochondrial transport [63]. A recent study used these techniques to mimic the role of myotubular contractility on axonal regeneration, where axons from motor neurons co-cultured with myotubes were injured using fluidic aspiration along with bubble introduction [64].

### 2.4. Tear on Cell Monolayer

While spatial separation of the cell body and injured axons closely mimics nerve injury in the PNS, a few studies have taken a more direct approach in inflicting neuronal injury. Mixed cultures of glia and neurons with rich axonal growth were injured by running a pipette tip diametrically along the length of the culture plate, resulting in a ‘tear’ measuring approximately 0.6 to 1 mm wide across the cultured cell layer [65]. A second tear was applied perpendicular to the primary tear, thereby yielding four quadrants within the cell culture plate. This model yields areas that contain undisturbed cells deep within the quadrants that are distant to the injury and also cells that are in close proximity to the injury site. Within 2–3 min of tear application, the neuronal soma adjacent to the tear demonstrated swelling and granularity [65,66]. Swelling was also observed in neurons located 1–2 mm away from the in vitro lesion. Shortly thereafter, neurons located much deeper within the quadrants developed features of swelling, while glial cells remained unchanged [65].

### 2.5. Laser-Induced Transection

To address the complexities of neurite injuries at the level of single cells, studies have utilized strategies that make use of laser-induced microsurgical lesions. Importantly, axonal transections induced by a focused laser are quite precise, with minimal off-target injury and cytoplasmic damage to cells present in the vicinity. Recently, cultures of neuronal and glial cells exposed to multiple 6-nanosecond laser pulses at 355 nm exhibited immediate partial retraction of axons from the site of laser-induced injury, followed by axonal beading and proximal segment retraction after 1 s [67]. Interestingly, cell bodies of neurons that were subjected to laser-induced axonal injury demonstrated numerous vacuoles [68], especially in the mitochondria [69]. Similarly, neurite injury induced by a UV laser beam resulted in proximal and distal segment retraction with signs of degeneration comparatively higher in thicker segments than thinner segments [68,70]. In another laser-based transection model, low-energy picosecond laser pulses were used to induce single axon transection in vitro, with micro-level resolution [71] (Figure 3). To maximize precision and accuracy, axonal injuries were performed on an inverted microscope coupled with a high-resolution motorized stage and an environmental chamber. Similar to single axon injuries observed in other laser-injury models, the low-energy laser pulses induced axonal swelling, distal segment degeneration, axonal retraction, and vesicle formation [67,71,72]. These observations lend strong credence to using lasers for inducing axonal transection in vitro to mimic membranal and cytoplasmic changes that occur in injured neurons in vivo.

However, it should be noted that an inherent drawback of the in vitro technique, no matter the model system used, lies with the isolation of neural tissue, such as the dorsal root ganglia [57] and the sympathetic ganglia that are commonly used in PNI. Isolation of ganglia for neuronal dissociation involves transection of the nerve roots, which itself causes several injury-induced cellular responses within cultured neurons [73]. Hence, it is possible that many experimental in vitro injury-induced responses in neurons may be masked by the initial injury caused during ganglia harvesting and isolation. Investigators need to be aware of this confounding variable before venturing into studies involving the in vitro model systems and design their experiments accordingly.

In summary, the main advantages and disadvantages of the transection model are listed below:

### 2.6. Advantages

Compartmentalization of peripheral neuron cultures provides robust separation of axon and soma that morphologically reflects a more in vivo arrangement.The spatial separation also allows for the application of consistent injury events, while the cellular response can be easily visualized and quantified using conventional approaches.In addition to being cost-effective, this model also allows for the performance of various biochemical assays that require higher volumes of injured cells.The use of laser-induced transection provides precision and accuracy for single-cell studies.

### 2.7. Disadvantages

Though the compartmental models are quite easy to set up, these devices are prone to fluid leak that may warrant repetition of experiments.There may be difficulty in the assembly of the device for advanced live-cell microscopy, and necessary modifications may have to be incorporated.Single-cell studies using laser-induced injuries will need advanced expertise for set-up and may not be possible in a typical laboratory setting.It is quite challenging to limit glial cells within compartments, and eventually, they tend to migrate into all compartments.

## 3. Axonal Stretch Injury Model

Traction on the peripheral nerve is a common occurrence due to trauma, surgical procedures, and limb lengthening maneuvers. Interestingly, the stretching force exerted on a nerve has been found to be proportional to the ensuing nerve ischemia, with at least 15% nerve elongation required for precipitation of ischemia [74] and up to 50% elongation for the appearance of histological changes [75,76]. Nerve tissue scarring and tearing of the nerve outer layers (epineurium, perineurium) along with the attached blood vessels are some of the commonly observed lesions in nerve-stretch injury [77].

Unlike in an axotomy, where injury-induced damage is all-or-none, stretch-induced injury demonstrates a range of nerve responses, including an axonal growth phase with mild stretch [78,79] and an injury phase above the resting tension threshold [79,80]. One of the first in vitro models designed to understand stretch-induced injury at the cellular level was in CNS neurons cultured on flexible silastic membranes [81,82], while similar methods have been extended to peripheral neurons as well. In these models, neurons cultured on a flexible silastic membrane are exposed to controlled stretching forces, mainly by the application of calibrated pressure pulses that results in global deformation of the silastic membrane, which in turn stretches cells that are adhered to it [83,84,85]. The actual stretching of the membrane may be as brief as 250 ms, but this is sufficient to produce cell deformation [81,82,83,86] (Figure 4). Further, low levels of stretch were found to induce changes in membrane potential along with delayed depolarization, without any significant alteration in cell viability [86]. Interestingly, membrane-stretched cells have been found to depolarize only after stretch release [83].

Although few studies have reported only mild effects on low levels of axonal stretch, there have been other studies that have observed stretch-induced structural abnormalities in mitochondria, and vacuolization of golgi and rough-surfaced endoplasmic reticulum [84,87,88]. In addition, neurons cultured on flexible polymeric substrates subjected to substrate stretching show a strain-rate-dependent decrease in fast axonal transport [89] and immediate hyperpolarization [83].

Another significant advancement in studying axonal injury is the adoption of microfluidic devices. Neurons cultured within microfluidic devices have been subjected to axonal injury by simply pushing fluid into intersecting microgrooves within the device [54,59,62,90]. Axonal stretching using a cantilever attached to a piezoelectric drive is another innovative approach to produce stretch-induced injury changes at the single axon level (Figure 5). Such in vitro models have contributed quite well to our understanding of the biochemical and gene expression changes in the cell soma at an individual neuron level.

It is also interesting to note that there are conflicting reports in terms of injury severity in mild stretch models [91], with some studies demonstrating beneficial effects of mild nerve stretch [92,93]. In contrast, severe axonal stretching induces critical and permanent deficits, including mitochondrial swelling and cell death [94,95,96]. However, what needs to be noted is that unlike single axons, bundles of axons seem to be quite resilient towards external strain. This could be because of axon–axon adhesions that increase the tensile strength of the whole bundle or due to bundling itself [97]. This makes the conclusions from single axon studies difficult to interpret, especially when trying to mimic responses related to strain rates in vivo.

In summary, the main advantages and disadvantages of the cell stretch model are listed below:

### 3.1. Advantages

Direct and real-time measurement of controlled mechanical strain.The cellular response is reproducible, and the principles behind stretching mechanisms are quite simple.

### 3.2. Disadvantages

Difficulty in mimicking various injury dynamics as seen in real-life conditions, where nerves may be exposed to stretching and twisting force.Techniques such as incorporating a PDMS membrane within microfluidic systems requires experience and advanced skills.Sum total of tensile strength of individual axons, being lower than the tensile strength of an entire nerve, makes in vitro models less relatable to in vivo observations.

## 4. Compression Induced-Injury Models

Compressive neuropathies are conditions that are associated with the physical compression of a nerve, resulting in pain, numbness, and possible loss of function [98,99]. Nerve compression can be either acute, as in traumatic crush injuries, or chronic compression, as observed in vertebral lesions and entrapment syndromes [100,101]. Few in vitro models have been developed to address the electrophysiological and morphological features of compression injury in nerves. Controlled compression applied on excised canine lumbar roots, with a rod equipped with a pressure gauge, and operated by a pulse motor, demonstrated considerable lowering of the threshold for ectopic firing and prolonged firing in DRG neurons [102]. Significantly, these in vitro findings were later validated in chronically compressed DRG neurons in vivo [103,104]. In vitro compression models have also been used to demonstrate altered electrogenic responses in sensory neurons and a likely causal link to inflammatory mediators [105,106], and gene transcription changes [107,108] have been suggested. Encouragingly, such findings have been validated in vivo and even associated with the development of chronic pain conditions in nerve compression events [109,110].

In vitro compression chambers are generally custom-built and typically face the challenge of maintaining sufficient gas exchange during pressurization procedures [111]. To circumvent this challenge, compression systems have been designed with a gas–liquid interface, where a continuous gas flow can be maintained to regulate hydrostatic pressure within the system [112]. Further, compression experiments performed on myelinated DRG-Schwann cell co-cultures with regulated gas exchange have demonstrated axonal degeneration and reduced axon numbers at pressures around 0.008 MPa [112]. Similar compression devices at higher pressures of 0.5 MPa in cultured sensory neurons have demonstrated the activation of apoptotic pathways, cytoskeletal abnormalities, disruption of the cell cycle, and the precipitation of necrosis [106,113].

### 4.1. Advantages

Reasonable similarity of in-vitro-induced cellular changes compared to in vivo experiments.

### 4.2. Disadvantages

Model offers limited insight into the biomechanics of the nerve injury because of the difficulty in measuring the strain produced at the site of injury and the rate of application of the strain.Since the biomechanical force of injury varies, it is difficult to predict and control the lesions formed and control which axons are affected and which are spared.

## 5. Hydrostatic Pressure Models

Microenvironmental tissue hydrostatic pressure (HP) is an important cellular cue that plays a critical role in regulating cell growth, differentiation, migration, and apoptosis, both in vivo and in vitro [114]. In particular, HP is involved in maintaining homeostasis during the development of the optic system, central nervous system, cardiovascular system, cartilage, and bladder tissue. In the CNS, tissue HP is generally between 5 and 15 mmHg [114] but differs substantially based on cell and tissue type. However, in the PNS, HP is approximately 3–5 mm Hg in the dorsal root ganglia and approximately 2–3 mm Hg in peripheral nerves [115]. The HP gradient in the peripheral nerve is maintained by the interstitial fluid within the endoneurial space (EHP), and any alterations in pressure are known to cause severe pathological effects, secondary to endoneurial ischemia, compression induced injury, and edema. These changes ultimately cause the deformation and collapse of blood vessels, release of osmotic solutes from nerve axons, and build-up of fluid at the proximal site of injury, causing severe damage at the injured nerve [116,117,118].

To mimic in vivo conditions and closely model the mechanical forces affecting the cellular microenvironment, various in vitro HP models have been developed where cells are subjected to HP changes via pressurized chambers, hydrostatic fluid columns, and use of a syringe pump system.

### 5.1. Pressurized Chambers

Pressurized chambers are widely used and typically contain 95% air and 5% CO_2_ fitted with automatic pressure gauges and controllers to monitor pCO_2_, pO_2_, and pH levels within culture systems [114]. Pressure changes are regulated by micro-balloons or custom-made mechanical pressure controllers that are designed to mimic physiological alteration in blood flow for an extended period of time [112,119]. To mimic chronic nerve compression, studies have been designed where DRG neuron-Schwann cell–fibroblast co-cultures were subjected to increased pressure of 40 kPa for 24 h, resulting in decreased neuronal viability and the release of lactate dehydrogenase (LDH) [112]. Neurons subjected to a 7 d high-pressure environment at 18–46 mmHg demonstrated higher levels of LDH compared to cells that were exposed for only 1 day.

Similarly, to understand cytoskeletal changes due to HP forces acting on DRG cell bodies, a gas-pressurized system exerting 0.5 MPa has been shown to induce lower cell density, shorter neurites, and irregular-shaped cells. In addition, microtubule disruption, structural degradation, oxidative stress, apoptosis, and cell cycle arrest were also observed in some neurons [106]. Similar observations related to cellular apoptosis and oxidative stress induced by high HP have also been reported in PC-12 cells [120].

### 5.2. Hydrostatic Fluid Columns

The generation of HP using static columns of cell culture media with variable column height to create pressure is another innovative in vitro approach. With a change in column height, corresponding changes in the distribution of gases (pCO_2_, pO_2_), nutrition, and pH will vary and, thus, experiments can be designed to study the role of varying HP on cells [114]. The HP experienced by cultured cells within columns can be regulated by either placing cells on coverslips at the bottom of medium-containing vertical columns [121] or by circulating the medium via a pressure-controlled peristaltic pump [122]. The latter arrangement also supplies fresh medium and gas to the cells. This set-up has been tried on cortical astrocytes but can be easily extended to peripheral neurons as well [123].

### 5.3. Syringe Pump System

Another technique to achieve fluctuations in hydrostatic pressure with culture medium is via the syringe pump method. The pressure monitored by a gauge can deliver positive (compressive), negative (tensile), or cyclic HP to the cells [114]. Injury is via a syringe pump, which applies increased hydraulic pressure to the media present in the cell culture chamber. Although not widely used as an in vitro injury platform, it has been observed that the use of different flow rates, needle bore size, and viscosity of the suspension can greatly affect the response of cells in the injection site [124]. This methodology could be further developed to mimic cellular response to injections and changes after peripheral nerve blocks.

### 5.4. Advantages

Cells can be subjected to a wide variety of controlled hydrostatic pressure ranges.In vitro platform helps in understanding the underlying mechanisms of CNS injuries, glaucoma, and cardiovascular research.

### 5.5. Disadvantages

There may be design complexity with the necessity of a pump system in a standard cell culture set-up.Fluid flow may induce shear stress in cells.

## 6. Metabolic Nerve Injury Models

Metabolic disorders such as diabetes, Gaucher’s disease, and Krabbe disease have been well-known to precipitate progressive neuronal injury and disrupt normal nerve function [125,126,127]. In addition, impaired axonal regeneration is also a common feature that is observed in many metabolic disorders, primarily due to disruption of vascular inputs, inflammatory response, and glia–neuron communication [128]. In particular, damage to the peripheral nerve (peripheral neuropathy) is one of the most common complications observed in patients with long-standing diabetes mellitus. Current animal models of hyperglycemia have shed considerable light on the pathogenesis of neural complications in diabetes. Several in vitro studies have contributed strongly to our understanding of the peripheral neuron response to hyperglycemic conditions. Most studies addressing metabolic cellular injuries fundamentally follow a common principle where neuron and/or non-neuronal cells are exposed to in vitro media conditions that closely mimic specific components of the metabolic disorder in question. More commonly, sensory neurons and glial cells derived from murine models, PC-12 cells, and the SHSY5Y cell lines are utilized for in vitro metabolic injury experiments. In this segment, we review techniques utilized to demonstrate metabolic injury in cells exposed to hyperglycemic conditions, since diabetes is a commonly researched topic.

Exposure of differentiated PC-12 cells to high concentrations (>9 mg/mL) of glucose has been shown to induce the altered release of dopamine [129], enhance reactive oxygen species (ROS), and reduce cell viability and neurite extension [130,131,132]. Similarly, dissociated sensory neurons cultured in glucose concentrations of more than 60 mM showed an increased number of cells that were positive for the injury marker, ATF3 [133]. Interestingly, increased ATF3 positive cells were also observed in sensory neurons that were harvested from rats with experimentally induced hyperglycemia [133], suggesting a strong co-relation between in vivo and in vitro models. Along the same lines, in vitro models involving neurons cultured from animals with experimentally induced hyperglycemia have consistently demonstrated elevated levels of ROS, reduced neurite outgrowth [134], and aberrant calcium ion homeostasis [135].

Apart from sensory neurons, sympathetic neurons harvested from superior cervical ganglia (SCG) and celiac/superior mesenteric ganglia (CG/SMG) are some of the other peripheral neural tissues that are commonly utilized for exploring the effect of hyperglycemia in vitro [136]. It must also be noted that there have been several studies that have combined metabolic injury models with other PNI models, but the decision remains with the investigator to choose the appropriate combination of models to expand on their specific research questions.

### 6.1. Advantages

Inexpensive and easy to perform.Can be utilized to study the effect of hyperglycemia on specific cell types of the peripheral nervous system.

### 6.2. Disadvantages

Pathogenesis of diabetes-induced nerve injury goes beyond merely the hyperglycemia condition. Hence, an in vitro model of hyperglycemic growth media may not exactly mimic the multi-component involvement of metabolic disorders as observed in vivo.

## 7. Chemically Induced Nerve Injury Models

PNI induced by chemical interaction with neural components is commonly seen in patients under anti-cancer treatment [137], anti-microbials [137,138], and local anesthetics [139,140,141,142]. Chemotherapy-induced peripheral neuropathies (CIPN) are a set of clinical conditions observed in patients administered with chemotherapeutics. CIPN commonly occurs as an undesirable, dose-limiting effect of several chemotherapeutics such as cisplatin, paclitaxel [143,144], vinblastine, and vincristine [145], mainly due to chemically induced damage to axons and cell bodies of peripheral neurons. Though the underlying mechanism of CIPN is not completely known, axonal degeneration or ‘dying back’ is a well-recognized histopathological hallmark [27,146], and understanding the molecular mechanism behind such pathogenesis is crucial to develop therapeutic strategies. In vitro models, especially using dissociated DRG cultures, have been utilized to understand the key cellular features of CIPN. The dose-dependent loss of neurites, fragmentation of neurites, and enlargement of the neuronal cell body have been observed in DRG neurons exposed to bortezomib, cisplatin, eribulin, paclitaxel, and vincristine [147,148,149].

Interestingly, a few studies have also utilized compartmental models to specifically examine the chemically induced dying back effect of axons. Exposure of axons to target chemicals including paclitaxel and vincristine shows clear dysregulation in axonal growth followed by loss of electrical activity [150,151,152]. In addition, in vitro studies in Schwann cell-DRG co-cultures have also shown that anticancer drugs produce a disruption of myelin formation and induce the de-differentiation of Schwann cells to a more immature state, an observation that is consistent with in vivo findings [153].

As with chemotherapeutics-related cellular changes in vitro, neuro-toxic mechanisms associated with the activation of common signaling pathways have also been observed with other drugs, including local anesthetics, alcohol, and phenol derivatives. Toxicity-inducing cellular mechanisms include the activation of common signaling pathways linked to intrinsic caspases [154], Akt [155], and mitogen-activated protein kinase [156]. Notably, the exposure of cultured rat DRG neurons to increasing concentrations of lidocaine and bupivacaine has been shown to induce neuronal apoptosis, Schwann cell death [157,158,159], and mitochondrial depletion via caspase and p38MAPK signaling pathways [160,161].

### 7.1. Advantages

Effect of chemotherapeutics on non-neuronal cells and its subsequent effect on neurons can be studied separately.

### 7.2. Disadvantages

Caution to be exercised when analyzing results from CIPN in vitro models, mainly because cancer, being a multi-component disease, will invariably have unknown variables that may be missed.

## 8. Blast-Induced Injury Models

Explosion-related blast injuries are a major health issue in military medicine, with improvised explosive devices causing many casualties in human conflicts [162]. Explosions cause unusual injury patterns that may involve skin, muscle, internal organs, and nerves. Intriguingly, an actual blast-linked injury has been shown to have four distinct mechanisms. The initial intense high-pressure impulse, referred to as a blast wave, is known to induce direct soft-tissue injuries, while the secondary, tertiary, and quaternary injuries relate more to injuries due to flying debris, physical displacement, and other events, respectively [163,164,165].

There have been few in vitro studies that have attempted to mimic the effect of the blast wave on peripheral neurons. One such study exposed DRG cultures enclosed within a rubber glove to the impact of a high-energy missile within a water-filled tube [166]. The impact caused high-frequency oscillating pressure waves within the rubber glove that somewhat resembled the blast wave seen during explosions. Interestingly, this study found that within 6 h of impact, almost all DRG neurons demonstrated cytoskeletal disruption, plasma membrane dysfunction, and neurofilament tangles [166]. In another blast injury model that was closer to reality, PC-12 cells submerged in water were subjected to the effect of single and multiple primary blasts generated from research department explosives (RDX). Blast-exposed cells showed distinct axonal beading and decreased cell viability with multiple explosion exposure [167].

### 8.1. Advantages

Functional or biochemical measures can be taken after injury at multiple time points.

### 8.2. Disadvantages

Limitations in stimulating shock wave under special circumstances such as trenches, cabin, and underwater explosions.Experiments with compounds such as RDX may require additional regulatory approvals and specialized safety oversight.

## 9. Challenges and Future Perspectives

While in vitro model systems of nerve injury are promising due to their simplicity in experimental design, there are, however, two broad challenges that need to be addressed going forward: (a) the importance of the co-existence of neuronal and non-neuronal cells in models, and (b) the limitations of 2D culture systems. As firmly established in the past decade, the pathogenesis of nerve injury is not limited to neurons or glial cells alone. The surrounding non-neuronal cells are key components in determining the level of injury and regeneration [168,169,170]. Secondary injury or secondary damage results from crosstalk between neighboring cells and neurons, which can be either neuroprotective or destructive. For instance, immune cells play a prominent role after initial nerve injury. Immune cells such as macrophages are known to mediate Wallerian degeneration and, hence, a co-culture system that can incorporate immune cells, glia, and neurons may provide more resemblance to an in vivo setting. However, the main challenge remains that it is difficult to maintain long-duration co-cultures of neuronal and non-neuronal cells since the proliferative rates of cells are quite different.

Secondly, cultures in a 2D platform have significant shortcomings. For instance, regeneration ability declines with a higher grade level of injury [100]. Grades III and IV, which include damage to endoneurial tubes along with axons, are known to disrupt the formation of aligned bands of Bungner, required for regeneration. Replicating Grade III and Grade IV types of injury is not possible in 2D culture models. The mechanical properties of neurons cultured in the 3D extracellular matrix considerably vary from 2D culture, which will invariably affect its response to experimental injury. Though microfluidic devices have made it possible to spatially separate the cell body and axon, somewhat mimicking an in vivo design, the 3D architecture is still lacking.

Organoids provide a platform to culture multiple cell types and develop 3D networks that are physiologically closer to tissues in situ. The field of organoid modeling in the peripheral nervous system is gradually developing, with the incorporation of neuro-immune and neuro-vascular cellular interactions. Though exciting, the field of organoid development for the PNS is still in its infancy, and exciting outcomes can be expected in the future

## 10. Conclusions

PNI coupled with poor nerve regeneration capabilities can be debilitating and compromises overall quality of life. A better understanding of the molecular mechanisms behind such pathogenesis can aid in developing innovative intervention strategies for repair and regeneration. A variety of in vitro models of PNI exist with their own advantages and disadvantages. We observe that the in vitro models are gradually evolving from simple 2D culture systems to complex 3D platforms, and this, we believe, will extend the field of in vitro studies and take us closer to breaking down the complex cogwheel of nerve injury.

## Figures and Tables

**Figure 1 ijms-23-00816-f001:**
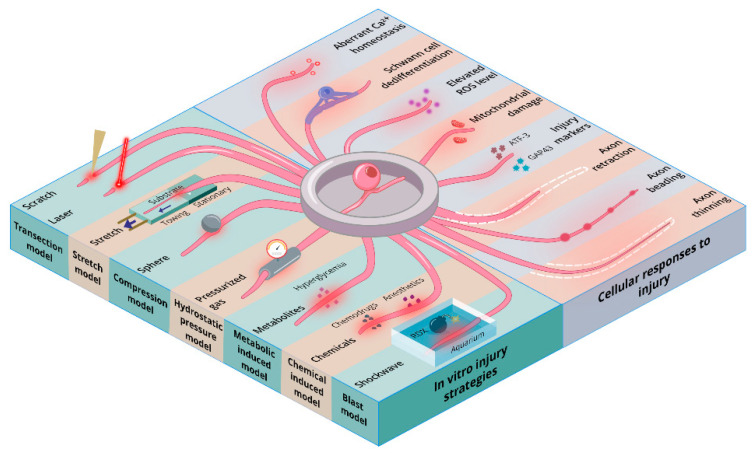
Schematic overview depicting various in vitro peripheral nerve injury models and peripheral cellular responses to injuries.

**Figure 2 ijms-23-00816-f002:**
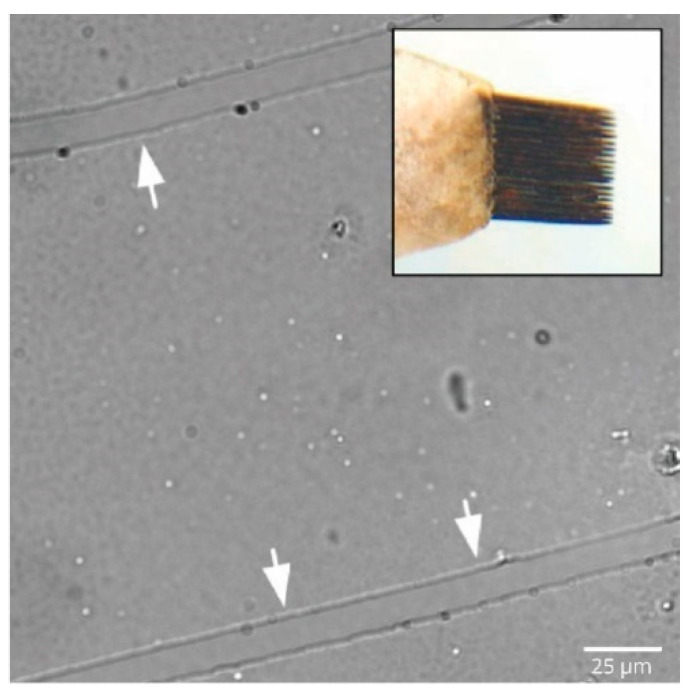
Aligned scratches on PDL-coated culture dish using rake pins for axonal guidance. Representative image of micron-sized scratches applied on poly-D-lysine-coated cell culture plates. Each scratch was approximately 14 μm in width and spaced approximately 150 μm apart. Inset shows the pin rake made by cementing together 20 individual insect pins, with each pin placed approximately 200 μm apart. Scale bar = 25 μm.

**Figure 3 ijms-23-00816-f003:**
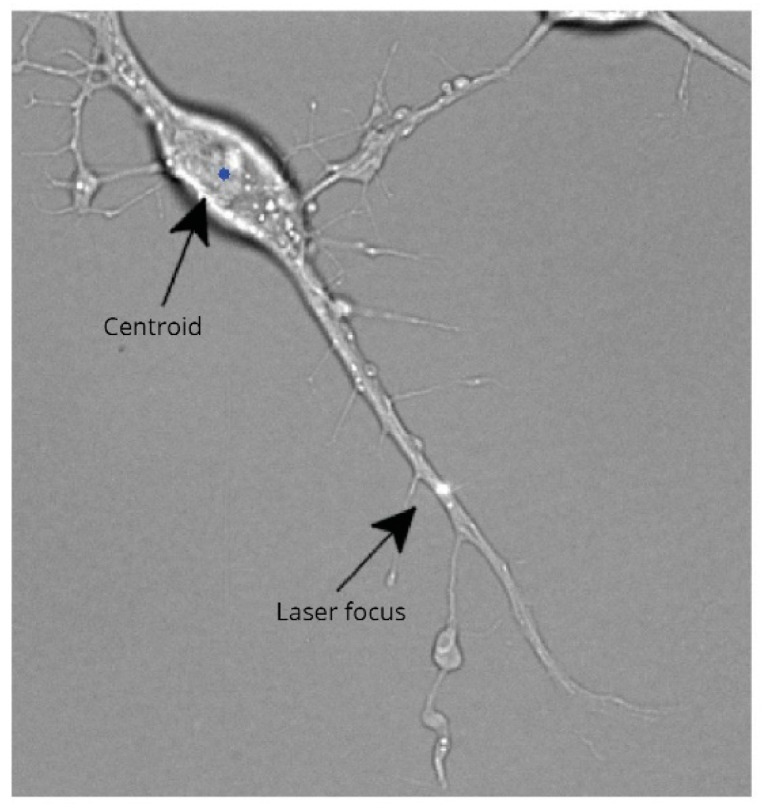
Laser-induced transection injury of an axon. Image showing a laser beam focused on neurite extension from a differentiated RGC-5 cell. Laser pulse was set to 7.5 nJ at 76 MHz, with exposure of 5 s (reproduced with permission from Santiago Costantino, Department of Ophthalmology, University of Montreal, Canada, PLoS ONE; 2011, 6(11), e26832.

**Figure 4 ijms-23-00816-f004:**
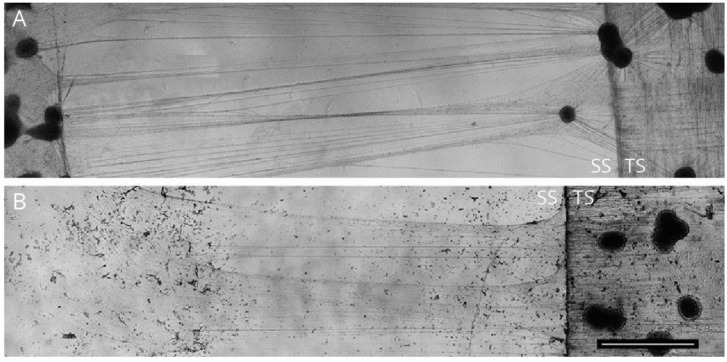
Axonal stretching from DRG explants. Image showing neurons from DRG explants cultured on a stationary substrate (SS) and a mobile towing substrate (TS). Neurons cultured on both SS and TS when exposed to controlled stretching force yielded axons with bi-directional polarity (**A**), while neurons cultured exclusively on TS yielded axons with unidirectional polarity (**B**). Scale bar = 2 mm (reproduced with permission from Bryan J. Pfister, Department of Biomedical Engineering, New Jersey Institute of Technology, Newark, NJ, USA. Journal of Neurotrauma, 28 (11), 2011, 2389–2403, Mary Ann Liebert, Inc.).

**Figure 5 ijms-23-00816-f005:**
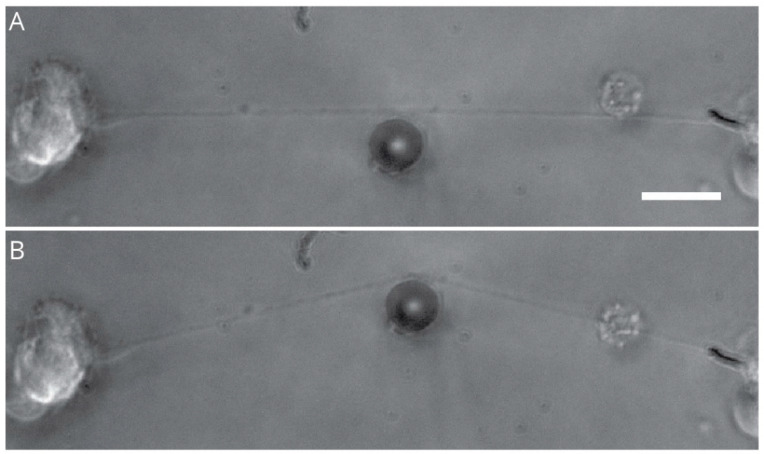
Stretching an individual axon. Image showing an unstretched (**A**) and a stretched (**B**) axon. The lateral stretching force was applied using an optical fiber cantilever. Scale bar: 20 μm (reproduced with permission from Pramod Pullarkat, Soft Condensed Matter Group, Raman Research Institute, Bengaluru, India, eLife; 2020, 9, 1–22).

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
