# Peer review of "A Brief Review of In Vitro Models for Injury and Regeneration in the Peripheral Nervous System"

_ijms, 2022, doi:10.3390/ijms23020816_

Round 1

Reviewer 1 Report

This review fills a gap in the literature. It summarizes, explains and highlights advantages and disadvantages of main in vitro peripheral nerve injury models.

The manuscript is well organized and clear to read. All important topics in the field are discussed. Figures are exhaustive. Literature cited is comprehensive of the main studies in the field.

Author Response

Reviewer 1:

Comment: This review fills a gap in the literature. It summarizes, explains and highlights advantages and disadvantages of main in vitro peripheral nerve injury models.

The manuscript is well organized and clear to read. All important topics in the field are discussed. Figures are exhaustive. Literature cited is comprehensive of the main studies in the field.

Response: We thank the referee for reviewing our article. Observations related to the work is appreciated. Thank you.

Reviewer 2 Report

Comment for authors

The study submitted for evaluation is a review of in -vitro models for injury and regeneration in the peripheral nervous system.

Recently, studies of model systems that mimic biological changes that occur during damage to human nerves have placed great emphasis on the research and development of in vitro models. In this context, any attempt to discuss the available techniques seems to be important, indicating examples of their specific use in research and a summary of their advantages and disadvantages. Therefore, the presented review is a valuable contribution to systematize our knowledge on the subject.

DETAILED REVIEW

The weakness of the reviewed work concerns the following issues:

  • The research presented includes current experimental techniques clearly described in the text. Nevertheless, selected topics were not covered in the manuscript. For this reason, I propose to change the title from "A Review of In -Vitro Models for Injury and Regeneration in the Peripheral Nervous System" to " A Brief Review of In -Vitro Models for Injury and Regeneration in the Peripheral Nervous System "
  • The presented review includes 161 citations, more than half of which covers papers published before 2017 (i.e. from the last five years), which appears to be very bad treatment of the reader as only 58 cited publications have been published since 2017. At the same time, the manuscript contains 10 citations published in the 1970s - 90s of the twentieth century. While only a cursory check of, for example, the PubMed database reveals about 1.500 works covering the thematic scope of the publication. I also believe that the topic has been developing vigorously over the past 5 years and focusing on old articles is not the best approach to reviewing it. Therefore, I am convinced that the authors must carry out a detailed literature review and supplement the citations with references to the latest research in this field.

I recommend publication after minor revision.

Author Response

Reviewer 2:

Comment 1: Recently, studies of model systems that mimic biological changes that occur during damage to human nerves have placed great emphasis on the research and development of in vitro models. In this context, any attempt to discuss the available techniques seems to be important, indicating examples of their specific use in research and a summary of their advantages and disadvantages. Therefore, the presented review is a valuable contribution to systematize our knowledge on the subject.

Response: We thank the reviewer for his/her time in reviewing this work and appreciate this observation.

Comment 2: The research presented includes current experimental techniques clearly described in the text. Nevertheless, selected topics were not covered in the manuscript. For this reason, I propose to change the title from "A Review of In -Vitro Models for Injury and Regeneration in the Peripheral Nervous System" to " A Brief Review of In -Vitro Models for Injury and Regeneration in the Peripheral Nervous System "

Response: We submit to the reviewer’s opinion that our review does not quite completely cover all issues related to neuronal injury and regeneration in in-vitro models. We have now changed the title as per the suggestion. We thank the reviewer for suggesting this modification.

Comment 3: The presented review includes 161 citations, more than half of which covers papers published before 2017 (i.e. from the last five years), which appears to be very bad treatment of the reader as only 58 cited publications have been published since 2017. At the same time, the manuscript contains 10 citations published in the 1970s - 90s of the twentieth century. While only a cursory check of, for example, the PubMed database reveals about 1.500 works covering the thematic scope of the publication. I also believe that the topic has been developing vigorously over the past 5 years and focusing on old articles is not the best approach to reviewing it. Therefore, I am convinced that the authors must carry out a detailed literature review and supplement the citations with references to the latest research in this field.

Response: We have revisited the references cited in the article and have updated references that were before 2017. Compared to the earlier manuscript version, we have now updated a total of 48 such references. However, we request that some of the historical references be retained, since it is important to provide background on how many of the in-vitro studies mentioned in the review were developed.

All changes in the revised article have been highlighted in yellow.